# High-Resolution, Integrated Hydrological Modeling of Climate Change Impacts on a Semi-Arid Urban Watershed in Niamey, Niger

**Boubacar Abdou Boko [1,2], Moussa Konaté [2], Nicaise Yalo [3], Steven J. Berg [4,5], Andre R. Erler [4,6], Pibgnina Bazié [7], Hyoun-Tae Hwang [4,5], Ousmane Seidou [8], Albachir Seydou Niandou [9], Keith Schimmel [9,*] and Edward A. Sudicky [4,5]**

[1] Graduate Research Program on Climate Change and Water Resources, West African Science Service Centre on Climate Change and Adapted Land Use (WASCAL), Université D'Abomey Calavi, Cotonou 03 BP 526, Benin; abdouboko@gmail.com

[2] Départment de Géologie, Faculté des Sciences et Techniques, Université Abdou Moumouni de Niamey, BP 10 662, Niamey 8000, Niger; konate.moussa@gmail.com

[3] National Water Institute, University of Abomey Calavi, Cotonou 03 BP 526, Benin; yalonicaise@yahoo.fr

[4] Aquanty Inc., 564 Weber Street North, Waterloo, ON N2L 5C6, Canada; sberg@aquanty.com (S.J.B.); AErler@aquanty.com (A.R.E.); hthwang@aquanty.com (H.-T.H.); esudicky@aquanty.com (E.A.S.)

[5] Department of Earth and Environmental Sciences, University of Waterloo, Waterloo, ON N2L 4C6, Canada

[6] Department of Physics, University of Toronto, Toronto, ON M5J 2S7, Canada

[7] AGRHYMET Regional Center, Niamey 8000, Niger; p.bazie@agrhymet.ne

[8] Department of Civil Engineering, University of Ottawa, Kingston, ON K7L 3R2, Canada; Ousmane.Seidou@uottawa.ca

[9] Applied Science and Technology, North Carolina Agricultural and Technical State University, Greensboro, NC 27411, USA; albachir@sahelconsulting.com

[*] Correspondence: schimmel@ncat.edu

**Abstract:** This study evaluates the impact of climate change on water resources in a large, semi-arid urban watershed located in the Niamey Republic of Niger, West Africa. The watershed was modeled using the fully integrated surface–subsurface HydroGeoSphere model at a high spatial resolution. Historical (1980–2005) and projected (2020–2050) climate scenarios, derived from the outputs of three regional climate models (RCMs) under the regional climate projection (RCP) 4.5 scenario, were statistically downscaled using the multiscale quantile mapping bias correction method. Results show that the bias correction method is optimum at daily and monthly scales, and increased RCM resolution does not improve the performance of the model. The three RCMs predicted increases of up to 1.6% in annual rainfall and of 1.58 °C for mean annual temperatures between the historical and projected periods. The durations of the minimum environmental flow (MEF) conditions, required to supply drinking and agricultural water, were found to be sensitive to changes in runoff resulting from climate change. MEF occurrences and durations are likely to be greater from 2020–2030, and then they will be reduced for the 2030–2050 statistical periods. All three RCMs consistently project a rise in groundwater table of more than 10 m in topographically high zones, where the groundwater table is deep, and an increase of 2 m in the shallow groundwater table.

**Keywords:** climate change; integrated hydrological model; semi-arid; impacts

## 1. Introduction

The Niger River is the primary surface water used for agriculture and the drinking water supply for Niamey, Niger, West Africa (located in the middle Niger River basin; see Figure 1a,b). Given that

the water distribution network does not cover the entire populated area, and because of recurrent droughts, the Niger River cannot fulfill the total water demand for the area. Groundwater is pumped through open wells and boreholes to provide water to more than 35% [1] of the city's population of 1.3 million [2].

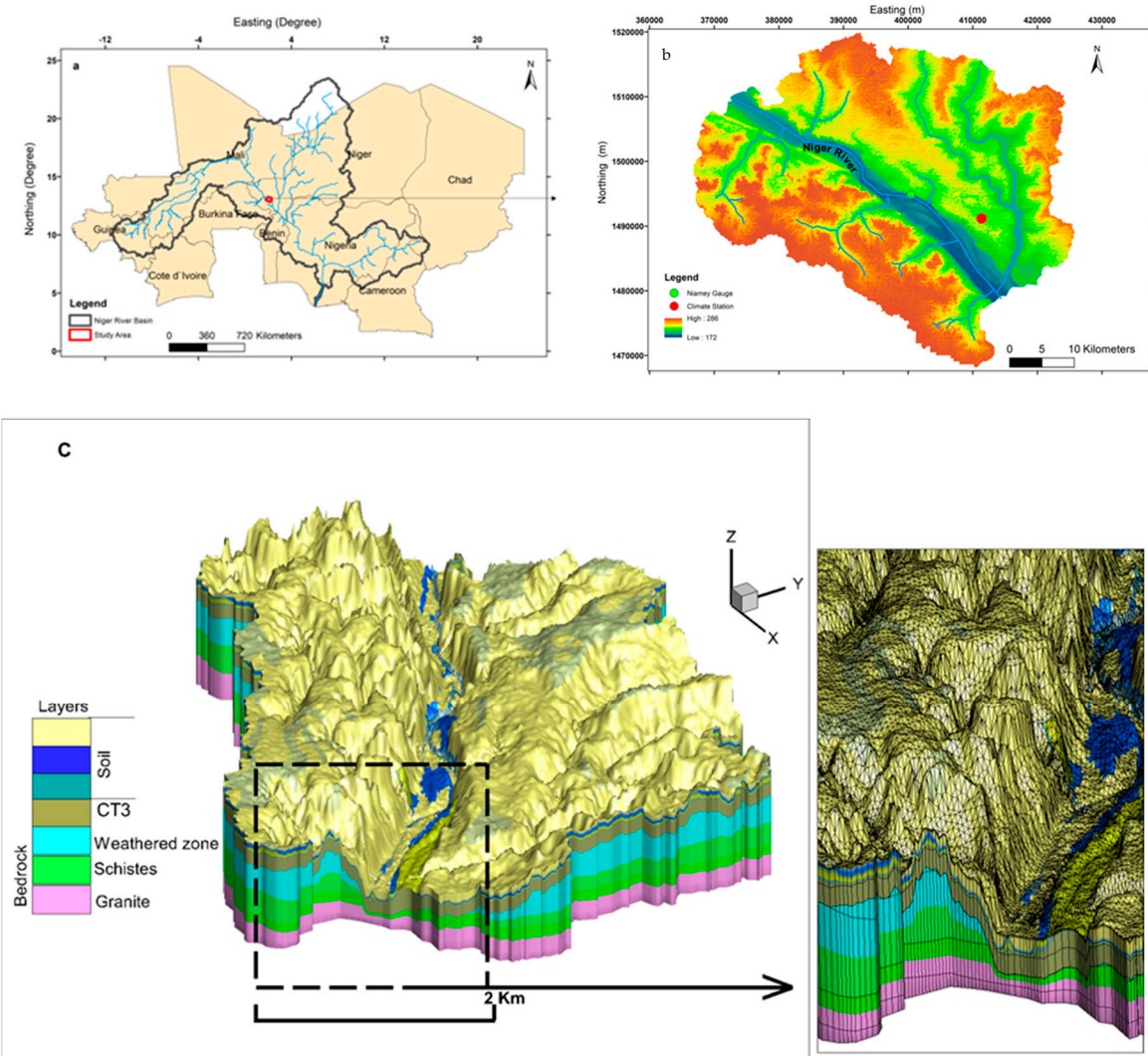

**Figure 1.** (**a**) Study area location within the Niger River basin. (**b**) Local context of the study area with topography and stream network. (**c**) Three-dimensional (3D) surface–subsurface model of the study area.

The demand on groundwater supply to meet drinking and agricultural water use needs is increasing, due to rapid population growth (doubling every 25 years) and urbanization. This is further exacerbated during low flow conditions in the Niger River, when there is a minimum environmental flow (MEF) target of 55 m$^3$/s. During these MEF periods, groundwater becomes the primary source of water. Additional stressors on the water supply include sedimentation of the riverbed and increased variability in streamflow upstream of Niamey. Therefore, the MEF for Niamey, set to 55 m$^3$/s over 10 days, is often violated. Authorities in Niger fear that groundwater resources may become insufficient or that violations of the MEF will become more severe and more frequent in the future as a result of climate change and increased demand. Therefore, a study that investigates the behavior of the whole hydrological system in the area under climate change is important, particularly in the context of the ongoing Kandadji dam construction upstream and the need to develop effective climate change mitigation strategies.

Understanding the impacts of climate change and hydrological extremes on water resources remains a central issue for sustainable water resource management. The impacts are likely to alter the hydrological cycle and induce negative effects on the availability and quality of water resources [3]. Changes in temperature and precipitation, combined with changes in the frequency and intensity of extreme hydro-meteorological events, will most likely have important implications for water resources by affecting the supply, quality, and distribution of water for billions of people [4]. However, the effects of climate change are not expected be distributed evenly among sectors, regions, communities, households, and individuals. Some of these are likely to be particularly vulnerable to changes in the global water system.

Even though simulations of the impact of climate change on water resources are highly uncertain [5–14], there is great confidence that African water resources systems are among the most vulnerable [15]. Uncertainties are particularly high when it comes to groundwater resources [16]. Most studies [16–21] have been focused on the climate change impacts on surface water, often neglecting groundwater, which is more complicated to model and assumed to be less vulnerable to climate change. Moreover, few [22–24] have considered large-scale, fully-integrated hydrological models when investigating climate change impacts.

While most arid regions rely on groundwater for agriculture and drinking water supplies [25], little attention has been given to climate change impacts on groundwater in Africa [16,17,26,27]. Most studies use saturated groundwater flow models or loosely coupled surface water–groundwater models to investigate climate change impacts on groundwater. Major issues in these studies include the estimation of aquifer recharge and the use of overly simplistic bias-correction methods for the projected climate time series.

Aquifer recharge is the most important parameter in the hydrological cycle, connecting groundwater directly to the atmosphere, and any error in its estimate leads to significant variability in the projected change in groundwater reserves. The groundwater recharge process in arid environments is mainly driven by localized recharge through surface water features [28–31]; this is not adequately represented in saturated groundwater flow models, where groundwater recharge is a user-specified parameter. In these models, there is a linear relationship between groundwater recharge and evapotranspiration, derived from climate forcing data.

It has been shown that biases are significant in most climate change impact studies on water resources [32]. These biases often determine the direction of climate change impacts on water resources. Therefore, a better estimation of evapotranspiration may lead to a better estimation of climate change impacts on water resources. This may be achieved through the use of fully-integrated models, capable of calculating actual evapotranspiration and integrating different recharge processes (focused recharge, direct recharge, ground water to surface water) considering land-use types [22–24,33]. In fully integrated hydrological models, groundwater recharge is no longer user-specified, but part of the solution provided by the model. There is no systematic linear relationship between groundwater recharge and climate forcing data, and evapotranspiration is computed internally and spatially, considering different land use, surface water features, and subsurface hydrostratigraphy.

This paper uses state-of-the-art high resolution, fully-integrated hydrological models, and multiscale statistical bias correction to investigate climate change impacts on groundwater resources in the Niger River basin. More specifically, the objectives of the paper are to evaluate (1) the potential climate change impacts on groundwater resources, and (2) the frequency and duration of the MEFs for the next 30 years, using state-of-the-art hydrological models and multiscale, statistically downscaled regional climate models (RCMs).

## 2. Materials and Methods

### 2.1. Study Area

The study area is a 1900 km$^2$ sub-watershed of the middle Niger River basin (Figure 1a) and is located southwest of the Republic of Niger. The Niger River is used to provide water for agricultural purposes and to meet the drinking water needs of 65% of the 1.3 million people in Niamey [1]. The remaining 35% is supplied by groundwater from two different aquifers: (1) the fractured aquifers of the Liptako basement, and (2) the sedimentary aquifer of the Continental terminal 3 (Figure 1c). The water-bearing formations of the fractured aquifer are Paleoproterozoic in age (2300–2000 Ma) [34], and are composed of granitoid plutons alternating with greenstone belts. The greenstone belts are composed of metamorphosed sandstone–pelitic rocks (shales, sericite schists, micaceous schists, quartzitic schists) and low-to-medium metamorphic greenstone (amphibolite, chloritoschists, metabasalts, metagabbros) [34–36]. The Continental Terminal aquifer is mainly composed of clayish to silty sandstones interbedded with oolithes and clay lenses, and overlays a major unconformity with the Liptako basement formation.

The topographic high within the study area corresponds to the CT3 plateau, with elevations ranging from 190 m to 250 m (a.s.l.), while the lowland area is occupied by plains containing ephemeral streams and ponds, with elevations ranging from between 178 m to 185 m (Figure 1b). Ephemeral streams act as temporary tributaries of the Niger River, and are depression-focused groundwater recharge areas [37]. Runoff collected by the ephemeral stream is generally discharged into several temporary or permanent ponds that are located in the course of the ephemeral streams. The ponds act as groundwater recharge/discharge areas during the rainy/dry season [37].

Climate in the area is semi-arid, and characterized by low-frequency, intense rainfall events occurring from June to September (rainy season). There is no rainfall in the long dry season, spanning from October to May (Figure 2). The rainfall and temperature (*Tmax*, *Tmin*) characteristics are shown in Figure 2, where D20, M3,Y1, and Y5 represent 20 day, 3 month, annual, and 5 year means calculated from daily averages. The mean annual rainfall for 1980–2009 was estimated as 514 mm, with a standard deviation of 116 mm [38], highlighting the important temporal rainfall interannual variability. The average temperature in the study area was 29 °C, while potential evapotranspiration was 2500 mm per year [39].

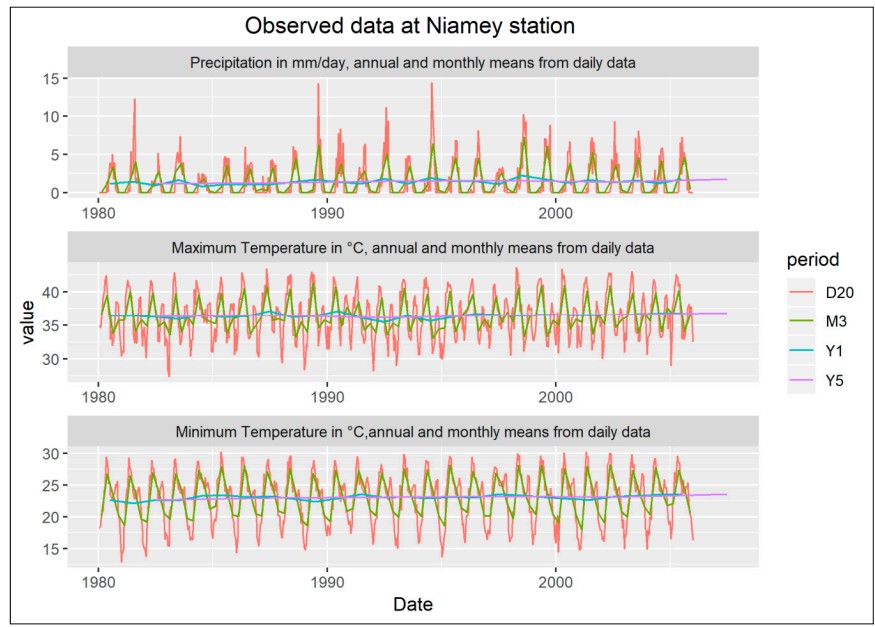

**Figure 2.** Observed rainfall, maximum temperatures, and minimum temperatures for 1980–2005 at Niamey Airport.

## 2.2. Integrated Hydrological Model

### 2.2.1. Mathematical and Numerical Model

The hydrological model used to simulate climate change impacts over the study area is the fully distributed, tightly-coupled, surface–subsurface, three-dimensional (3D), finite element model HydroGeoSphere (HGS) [40]. The 3D modified form of the Richards equation in the variably saturated subsurface flow domain and the two-dimensional (2D), depth-integrated diffusion wave equation for surface water flow are simultaneously solved in HGS in a parallelized manner [41].

One important source of uncertainty in climate change impact simulations is the calculation of evapotranspiration, a key connector between climate projections and the hydrological model [32,42]. The biases in evapotranspiration estimates can be reduced by describing evapotranspiration as a mechanistic process [32]. While evapotranspiration is a user-defined input in most studies, the evapotranspiration model used in HGS is a mechanistic process driven by the potential evapotranspiration and controlled by evaporation and transpiration parameters, such as soil moisture, surface water depth, land-use type, leaf area index (LAI), and rooting and evaporation depths [43]. Such a representation allows the reduction of biases in evapotranspiration biases using a mechanistic evapotranspiration model, as is done in climate models. Thus, the evapotranspiration model can be simplified based on the purpose of the study [44,45].

The evapotranspiration model used in this study is based on [43], and assumes that evaporation and transpiration affect both the surface and subsurface flow systems when evaporation is a result of energy penetrating the vegetation cover. Detailed information on the evapotranspiration implemented in HGS can be found in the HGS documentation [40]. The transpiration rate ($T_p$) is calculated using a function of a set of parameters that describe the net capacity of transpiration:

$$T_p = f_1(LAI)\, f_2(\theta)\, RDF \left( E_p - E_{can} \right) \tag{1}$$

where $f_1(LAI)$ is a function of leaf area index, $f_2(\theta)$ is a function of nodal water content, $RDF$ is the time-varying root distribution function, $E_p$ is the potential evapotranspiration rate, and $E_{can}$ is the canopy evaporation rate. The surface and subsurface evaporation ($E_S$) can be expressed as

$$E_s = \alpha^* \left( E_p - E_{can} \right) \left[ 1 - f_1(LAI) \right] EDF \tag{2}$$

where $\alpha^*$ is the wetness factor, based on surface water depths and subsurface soil saturations, and $EDF$ is the evaporation distribution function, which includes the surface/subsurface flow domains.

### 2.2.2. Discretization and Calibration

The 1900 km$^2$ area of the basin was discretized into triangular mesh elements, with a total of 516,901 nodes and 927,030 elements. The horizontal resolution of the 2D mesh elements ranged from 70 m at the vicinity of surface water features to 350 m in the rest of the model domain. The HGS model was sequentially calibrated with increasing temporal resolution at steady state, dynamic equilibrium, and fully transient conditions, as described in [37]. The outer subsurface model domain boundaries were assumed to correspond to groundwater flow divides, and no-flow boundary conditions were assigned. Precipitation and potential evapotranspiration were assigned to the top of the model as hydroclimate forcing variables. A critical depth condition was prescribed to the outer edge boundary of the model at the outlet of the watershed to let surface water flow out of the model domain. To represent the surface water flowrate generated upstream (outside) of the study area, a surface water flux boundary condition was assigned at the northernmost point. The discretization and calibration of the model, as well as the boundary conditions, are described in more detail in [37].

## 2.3. Hydroclimatic Data

### 2.3.1. Historical Climate Data

The model was forced with daily transient data of 25 years (1980–2005) of observed precipitation, along with maximum and minimum temperature (*Tmax*, *Tmin*) data from the Niamey Airport station (Figure 2). Potential evapotranspiration was calculated using the Hargreaves method:

$$E_p = 0.0023 \times 0.408 \, RA \times \left(T_{avg} + 17.8\right) \times TD^{0.5} \tag{3}$$

where *RA* is extraterrestrial radiation, expressed in MJ m$^{-2}$ d$^{-1}$; $T_{avg}$ is the average daily temperature (°C, defined as the average of the mean daily maximum and mean daily minimum temperatures); *TD* is the temperature range, estimated as the difference between mean daily maximum and mean daily minimum temperatures; and 0.408 corresponds to the constant used to convert the radiation to its evaporation equivalent in mm. The Hargreaves reference evapotranspiration method is recommended when there is not sufficient data to compute the Penman Montheith reference evapotranspiration [46].

### 2.3.2. Hydrological Data

The Niamey gauging station was used to calibrate the model for surface water flow using 25 years (1980–2005) of daily stream flow data provided by the Niger Basin Authority (NBA). Twenty-five groundwater observations were used for steady-state groundwater head calibration for 1980–2005. No transient groundwater head observation data are available for the 1980–2005 simulation period, and the model was previously calibrated for 2012–2017 with transient data [37]. While uncertainty related to groundwater models may be important between calibration and prediction periods, these uncertainties are considerably reduced for physically-based models, even with different climatic conditions between calibration and prediction periods [10]. Therefore, the physically-based HGS's simulated groundwater heads for 1980–2005 could be used to predict climate change impacts on groundwater, based on the HGS-calibrated simulation of 2012–2017, in the absence of observations with relatively fewer uncertainties.

### 2.3.3. Climate Projections

The outputs of two regional climates models of the CORDEX experiment at 50 km resolution and one high-resolution regional climate model (RCM) of the West African Science Service Center on Climate Change and Adapted Land Use (WASCAL) experiment at 12 km resolution were selected as climate projections. Only RCP (representative concentration pathway) 4.5 was used because of the limited computational resources available, and due to the fact that the differences between RCP 4.5 and RCP 8.5 are minimal before 2040 [47]. CANRCM4-CANESM2 and RCA4-IPSL-CM5A are the two RCMs selected from the CORDEX experiment, based on their ability to reproduce the hydrological cycle in the Niger River basin [48]. The metrics used to evaluate the models are well described in [48]. The WASCAL WRF-GFDELM-ESM2M at 12 km resolution was selected to evaluate the added value of the high resolution. Table 1 describes the configuration of the RCM runs, including forcing global climate models (GCMs) and RCM resolutions.

**Table 1.** Regional climate models (RCMs) and the forced global climate models (GCMs) used in this study.

| Institution | RCM | GCM | Resolution |
|---|---|---|---|
| Canadian Centre for Climate Modelling and Analysis | CanRCM4 | CanESM2 | 50 km |
| Institut Pierre-Simon Laplace, France | RCA4 | IPSL-CM5A | 50 km |
| West African Science Service Center on Climate Change and Adapted Land Use (WASCAL) | WRFV3.5.1 | GFDL-ESM2M | 12 km |

### 2.3.4. Bias Correction

Hydrological impacts of climate change are typically evaluated using dynamic or statistical bias-corrected climate output to force the hydrological models. Dynamic downscaling involves the use of physics-based RCMs with relatively higher resolution than the forcing GCM. Theoretically, the higher resolution allows a direct resolution of relevant local climate features. Dynamic downscaling is computationally expensive, hence its use in large-scale hydrological impact studies is limited. It is much easier to use statistical bias correction methods, which generate a mapping function between predictor fields derived from the observed local climate to local climate variables. The mapping function converts the simulated climate outputs into a corrected time series whose statistical properties are closer to that of the local observed climate data. The calibration of the mapping function is usually done for the historical or control period where observed data are available. The calibrated mapping function is then applied to the prediction period to obtain climate change projections.

However, a commonly used statistical bias correction method in hydrological climate impact studies assumes the mapping function to be valid at all temporal resolutions. Thus, it may introduce errors in impact studies [49] and a multiscale bias correction to allow different mapping functions at different scales. The method uses a standard quantile mapping method iteratively at multiple combined timescales (daily, monthly, and annual). To reduce potential errors associated with the stationary assumption, historical and projected climate output from the three RCMs were bias-corrected to the observed climate station at Niamey Airport (see Figure 2) using two different methods: (1) standard quantile mapping, calibrated on a daily timescale; and (2) multiscale bias correction, calibrated on daily (D1), monthly (M1), seasonal (M3), and annual (Y1) timescales. A comparison between the two methods was then performed to choose the best bias correction method. Statistical downscaling was applied to historical (1980–2005) and mid-century (2020–2050) periods for each climate model, resulting in a total of six (25 and 30 year) daily transient simulations.

The multiscale bias correction method developed by Hanel et al. [49] aims to reduce the error introduced by the calculation of the bias for a single scale by the majority of statistical bias correction methods. The method uses standard quantile mapping iteratively at multiple combined timescales (daily, monthly, and annual), as described in Equations (4) and (5) for precipitation and temperatures variables:

$$X^M_{s[0]} t(0) = X^C_{s[0]} \prod_{i>0} \frac{X^C_{s[i]} t(i)}{X^A_{s[i]} t(i)} \tag{4}$$

$$X^M_{s[0]} t(0) = X^C_{s[0]} \sum_{i>0} \frac{X^C_{s[i]} t(i)}{X^A_{s[i]} t(i)} \tag{5}$$

where $X^M_{s[0]}$ represents the variable to be corrected iteratively at different time scales, and $S = S[0], S[1], S[2] \ldots$, with $S[0]$ representing the original time scale. $X_{s[i]}$ corresponds to the time series in which $X$ is aggregated into $S[i]$ scales, and is independently corrected for every time scale using the standard quantile mapping bias correction method. Therefore, $X^C_{s[i]}$ represents the resulting precipitation time series corrected at multiple time scales. Values at different time scales, $S[i]$, can be found by aggregating the corrected variables from the closest smaller time scale, such as $X^A_{s[i]} = A\left(X^C_{s[i-1]}\right)$ with $A$ equal to the sum from $S[i-1]$ to $S[i]$, and the elements of the time series at the origin are specified by temporal indices corresponding to $t_{s[O]}, t_{s[1]}, t_{s[2]} \ldots$ .

## 3. Results and Discussion

In this section, the biases of uncorrected and corrected RCM simulations of precipitation and evapotranspiration are examined, and simulated groundwater heads and depths to groundwater, as simulated by the HGS model, are compared to observations. The projections of the HGS model in terms of groundwater recharge and MEFs are presented and discussed.

### 3.1. Biases in Uncorrected and Corrected Historical Climate Simulations

The observed historical (1980–2005) rainfall and temperature data at Niamey airport station (Figure 2) are compared at different timescales in Figure 3, with the basin-weighted, average, uncorrected historical precipitation simulated by the three RCM models. Rainfall biases are plotted as relative differences between historical observed and simulated data, while biases in temperatures are calculated as absolute differences averaged over different seasons. In Figure 3, DJF is the December–January–February season; JJA corresponds to June–July–August, and is the rainy season; and MAM and SON are the March–April–May and September–October–November periods.

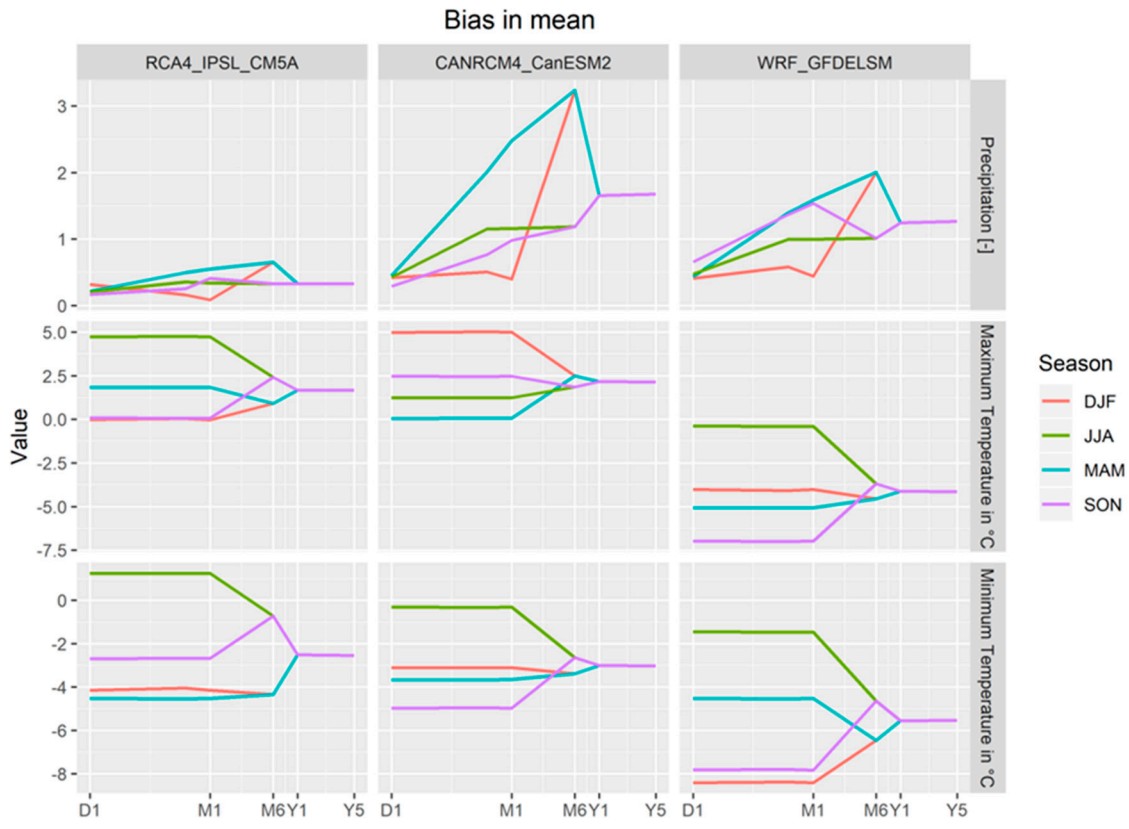

**Figure 3.** Bias in the mean basin average precipitation and temperature at different timescales between observed and simulated climate model data for the historical period 1980–2005.

Considerable differences in the statistical characteristics of the mean biases at different timescales can be seen in Figure 3. The biases increase as time intervals increase from daily to monthly, sub-seasonal, and seasonal (yearly) averages. CANRCM4 and WRF present the largest biases in mean precipitation, while the RCA4 has less bias in rainfall and temperature data. All three regional models perform better during the JJA period, which corresponds to the rainy season, confirming the ability of the selected models to reproduce the seasonal cycle of precipitation [47,48]. All three RCMs have large positive biases in mean simulated historical rainfall compared to observed historical rainfall. Rainfall biases are greater for the CANRCM4 model, followed by the WRF and RCA4 models, and particularly for

the MAM period, where the biases in the mean reached up to 300%. The observed large biases are probably due to the general wet-day biases of uncorrected RCMs.

As for the temperatures, the CANRCM4 and RCA4 models show positive maximum temperatures biases, whereas the WRF model is negatively biased for all seasons. The three RCMs have negative minimum temperatures biases for all the seasons, except for the JJA period, where they are positively biased, with the WRF recording the largest minimum temperature biases. Biases calculated using corrected and original (uncorrected) rainfall, along with maximum and minimum temperature time series for the three RCMs, are shown in Figure 4a–c for the standard and multiscale quantile–quantile mapping bias correction (QQ_BC) methods, and for different seasons of the year. The observed historical climate data (see Figure 2) is used to evaluate the performance of the two bias correction methods for the three RCMs for the simulated historical period (1980–2005). In Figure 4a–c, residual biases are averaged and expressed as relative difference values (to observed historical) for rainfall data and in absolute differences for temperatures data on daily (D1), monthly (M1), and yearly (Y1) timescales.

The bias in rainfall, calculated with the uncorrected WRF historical data (Figure 4a), varies from 50% to more than 250% across different periods, with large biases recorded for the SON period. Simulated rainfall data for the RCA4 model are relatively less biased, with the MAM period recording the largest bias of about 100% (Figure 4b). The CANRCM4 model has relative rainfall biases ranging from less than 100% to more than 450%, with the MAM period having the largest bias (Figure 4c). The mean rainfall biases increase from daily to monthly temporal scales, then decrease on a yearly timescale for the WRF and RCA4 models. In contrast, the biases are larger at smaller temporal scales for the CANRCM4 model. Absolute temperatures biases show different patterns for the three RCMs across different temporal scales and periods. *Tmin* and *Tmax* simulated by the WRF model are negatively biased, with differences of up to −6 °C for *Tmax* and up to −8 °C for *Tmin* in the JJA period (Figure 4a). *Tmax* has positive biases in both RCA4 (Figure 4b) and CANRCM4 (Figure 4c) simulations across all periods, except for the SON period for the RCA4 model, which has a slightly negative *Tmax* bias. *Tmin* in the RCA4 and CANRCM is negatively biased across all the periods, except for the JJA period of the RCA4, which has both positive and negative *Tmin* biases.

The residual rainfall bias across the whole range of temporal scales is significantly reduced (less than 100%) by the standard quantile bias correction methods for all the periods. The multiscale bias correction method produced the same performance for the CANRCM4 model. For a given model, the standard bias correction method was more efficient on the daily timescale, while biases remain large at the monthly timescale (up to 2%). The multiscale bias correction method eliminated the residual temperatures biases across all the timescales for all periods.

As for the WRF and RCA4 models, the standard bias correction slightly reduced the rainfall residual bias on a daily (D1) timescale and increased the biases on a monthly (M1) timescale. The multiscale bias correction method removed significant bias at all the temporal scales. For both models, temperatures biases were completely removed by the multiscale bias correction method at all temporal scales, while in the standard method temperature biases were still important on monthly (M1) timescales. In general, both bias correction methods performed better for temperature than for rainfall. The standard quantile mapping method has proven to be more efficient on the daily timescale (D1) than on the monthly timescale (M1), while the multiscale method performed well across all temporal scales considered. Therefore, even though the WRF model has a higher spatial resolution (12 km) compared to the CANRCM4 and RCA4 models (50 km), it does not improve the model's performance. Statistical downscaling appears to be necessary when using such biased models in impact studies.

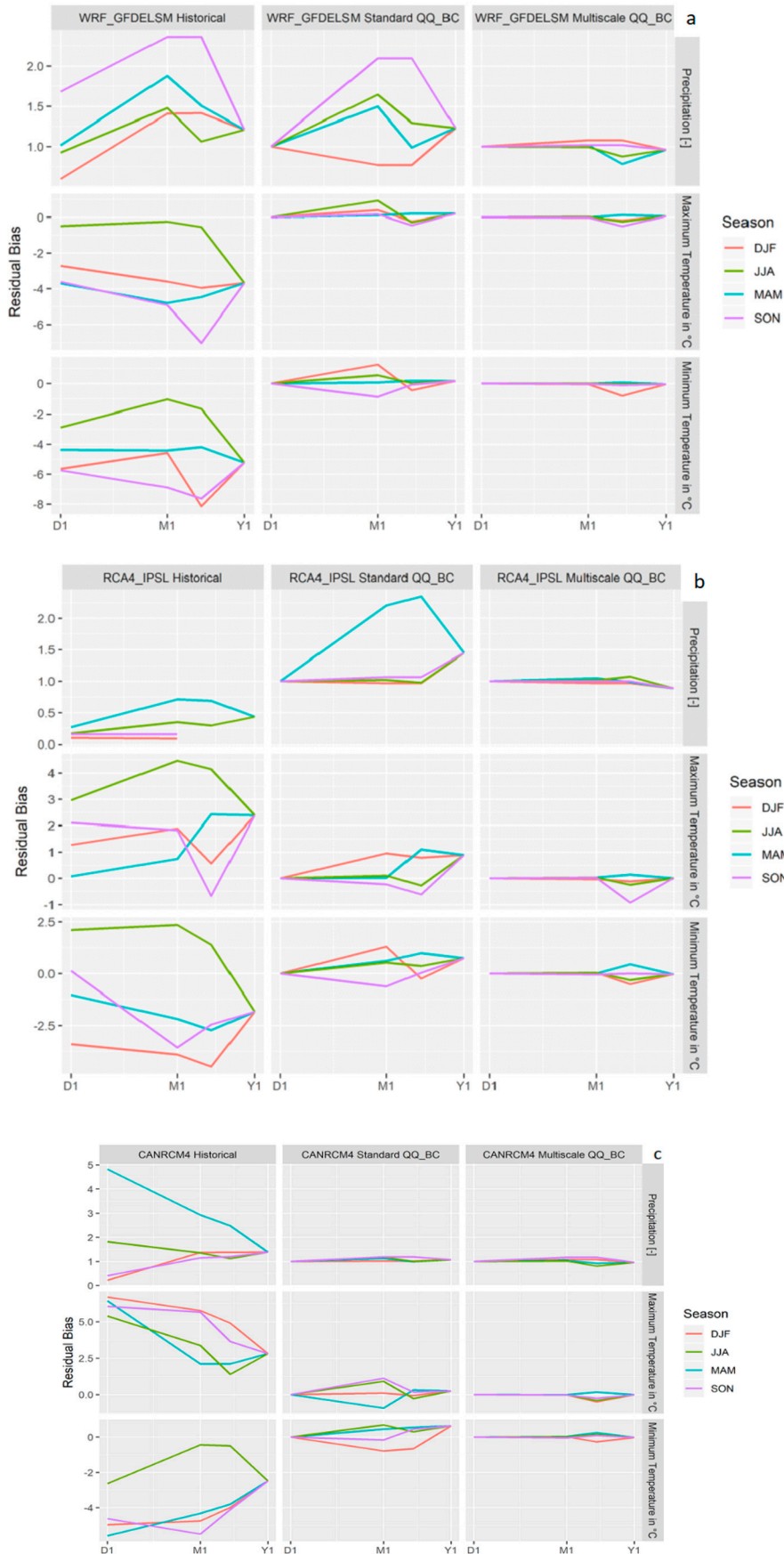

**Figure 4.** Standard quantile mapping and multiscale bias correction for the three RCMs: (**a**) WRF, (**b**) RCA4, and (**c**) CANRCM4.

*3.2. Validation of Historical Simulations against Observed Groundwater Levels*

It was shown in the previous section that even the statistically downscaled historical climate simulation data used to force the hydrological model still have substantial biases; therefore, a validation of historical hydrological simulations is necessary before conducting any impact studies driven by these climate forcing data. The depth-to-groundwater table and groundwater heads will be used as metrics for validation of the subsurface component of the HGS model. Depth to groundwater is a variable of great interest for water resource managers in the study area. It is crucial for drilling and managing water supply wells, for both drinking and agricultural purposes. The depth-to-groundwater table elevation is calculated in HGS as a linear interpolation of the pressure head at a null pressure level. A depth-to-groundwater table is then derived from subtraction between the elevation of the groundwater table and surface elevations calculated from the digital elevation model (DEM).

The observed and simulated depth-to-groundwater tables corresponding to the historical period 1980–2005 are mapped in Figure 5. Four groundwater observation well locations are plotted in Figure 5, and they will be used in the next section to show the long-term, simulated, transient groundwater head trend (Figure 6). Across the study area, the average depth-to-groundwater table ranges from less than 5 m to 65 m (Figure 5). Small depth-to-groundwater tables are generally along the low topographic areas that coincide with either the Niger River or ephemeral streams and ponds, where important exchange flux processes between surface water and groundwater occur [37]. Depth-to-groundwater tables are greater near high land areas and at many piezometric domes, as a result of important topographic control over the groundwater flow system in the study area, as shown in previous studies [37,50].

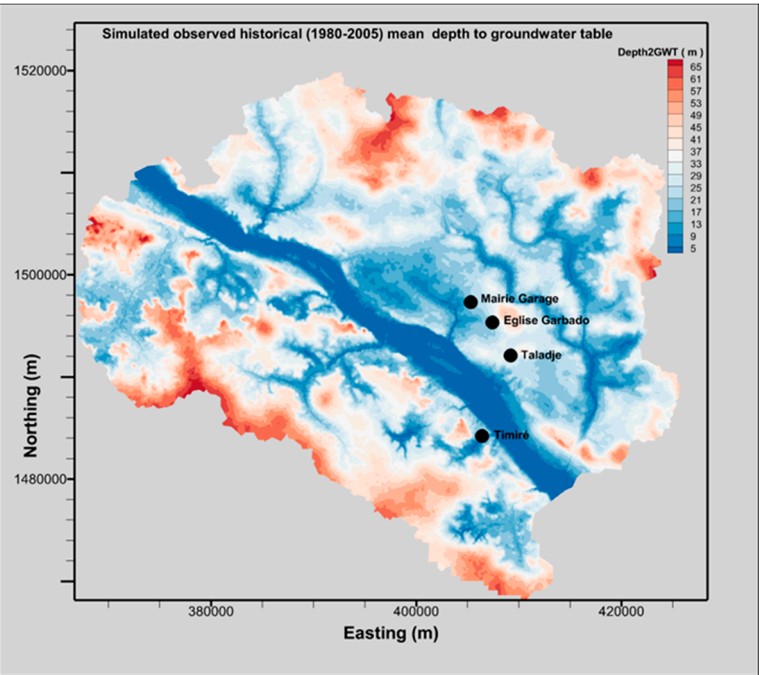

**Figure 5.** Simulated observed historical mean depth-to-groundwater table for 1980–2005.

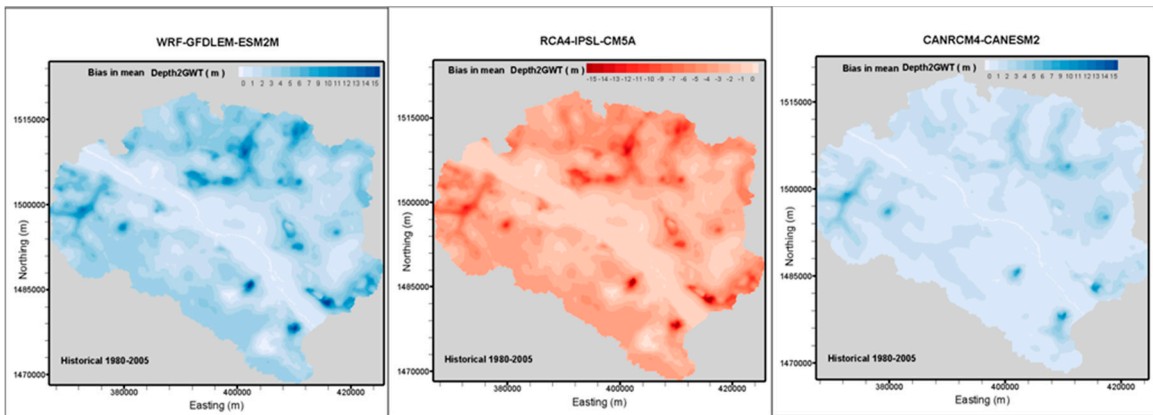

**Figure 6.** Bias in mean depth-to-groundwater table.

To validate historical simulations of depth-to-groundwater tables for the three RCMs, the bias in the mean depth-to-groundwater table is shown in Figure 6. The bias in mean depth-to-groundwater table is calculated as the difference between simulated observed historical climate data and simulated historical climate scenarios for the three RCMs considered.

Simulations with the WRF and CANRCM4 models lead to a higher mean depth-to-groundwater table, while the RCA4 is negatively biased (Figure 6). The bias in mean depth-to-groundwater tables ranges between 0 and +15 m for WRF and CANRCM4, and between 0 and −15 m for RCA4. Therefore, simulated historical climate scenarios are drier for the WRF and CANRCM4 models and wetter for the RCA4 model compared to the simulated observed historical depth-to-groundwater table. This should be directly linked to the residual bias of rainfall introduced by the multiscale bias correction method (see Figure 4a–c), where RCA4 still has greater positive rainfall bias compared to WRF and CANRCM4 for the JJA (rainy season) period.

The bias in mean depth-to-groundwater tables is spatially different in the study area with a high topographic area, having more bias than the lower zones. The effect of topography on the bias will be discussed thoroughly in Section 4. However, it is still important to validate simulations against the long-term transient groundwater head, in orders to better analyze how historical climate scenario performs reproduce long-term seasonal groundwater head variation. A time series of simulated groundwater heads at four observation wells (see Figure 5 for locations) are shown in Figure 7. Table 2 provides information on well altitudes and distances between wells, as well as simulated observed historical and simulated climate scenario heads. The four observation wells were chosen because they have no groundwater pumping and show the important topographical perturbation on groundwater response to climate change. They also have some historical groundwater head measurement records.

**Table 2.** Groundwater observation wells with the aquifer types, altitudes, distances between wells, and groundwater heads for the three RCMs.

| Well Name | Aquifer Type | Altitude (m) | Simulated Observed Historical Heads (m.a.s.l.) | Distance (km) | CANRCM4 (m.a.s.l.) | RCA4 (m.a.s.l.) | WRF (m.a.s.l.) |
|---|---|---|---|---|---|---|---|
| Mairie Garage | Fractured Granites | 220.6 | 206.0 | | 206.7 | 206.8 | 207.4 |
| Eglise Garbado | Fractured Schistes | 221.7 | 201.8 | 3.2 | 202.7 | 203.8 | 205.0 |
| Taladje | Fractured Quartzite | 225.0 | 199.4 | 3.5 | 200.1 | 201.1 | 202.3 |
| Timire | CT3 | 210.0 | 189.2 | 7.0 | 189.7 | 189.5 | 189.9 |

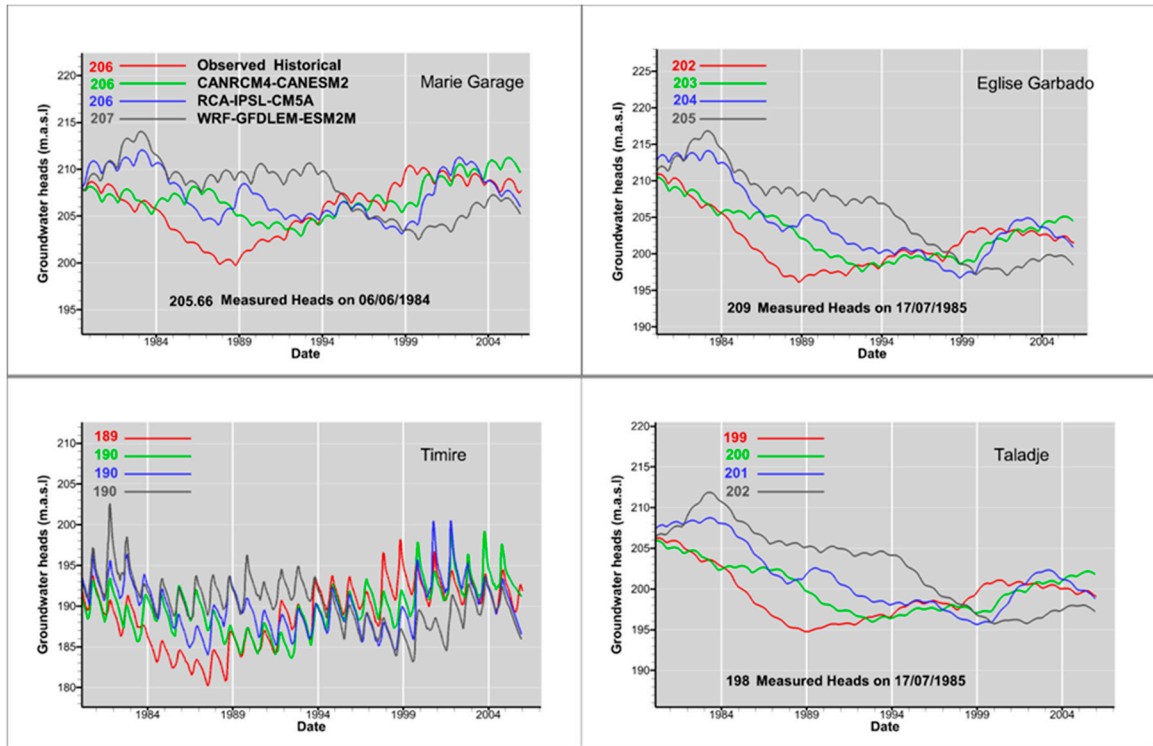

**Figure 7.** Simulated historical long-term (1980–2005) groundwater heads for the wells described in Table 2.

Figure 7 shows that the historical simulated groundwater heads from the climate scenario match reasonably well with the long-term seasonal variability of the observed simulated historical groundwater heads. Average 1980–2005 groundwater heads are shown as values, and groundwater head measurements are plotted at the bottom of each figure. These measured groundwater heads were reconstituted from historical measurements performed during the 1980s at the construction of the boreholes (Figure 7). All three RCMs tend to overestimate the groundwater heads for 1980–1994 and underestimate heads for 1994–2005. WRF shows the largest dry (wet) groundwater head bias, and CANRCM4 has the smallest dry (wet) head bias, while the CRA4 lies in between them (Figure 7). Historical transient groundwater heads, as simulated herein, show a general decrease between 1980–1994 and an increase from 1994–2005. This long-term increase of groundwater heads is probably due to the recent Sahelian rainfall regime changes recorded in the central part of the Sahel [51], where the 1989–2007 average rainfall exceeded the average rainfall for 1979–1990 by 10%. The increase in groundwater heads highlights the important sensitivity of the groundwater response to rainfall pattern changes.

As shown in Table 2, the mean error between simulated and observed groundwater heads seems to be greater at the observation wells where the depth-to-groundwater table is deep, and smaller at shallow groundwater tables. This topographic perturbation of groundwater response to climate change was recently shown in the Grand River watershed, Canada [22]. For [22], the topographic perturbation is not important within a few kilometers of horizontal resolution. However, Table 2 shows that in the study area, the topographic perturbation is still important, even for wells located within shorter distances. As soon as the altitude gradient exists, wells at high topographic positions are more biased than wells located in topographically low areas.

### 3.3. Validation of Historical Simulations against Surface Flowrate

Measured surface flow rates of the Niger River at the Niamey gauging station were used to validate the surface component of the integrated HGS model. Figure 8 shows the measured and

simulated historical flowrates for 1980–2005. Simulated flowrates match the measured flow well during all the simulated periods.

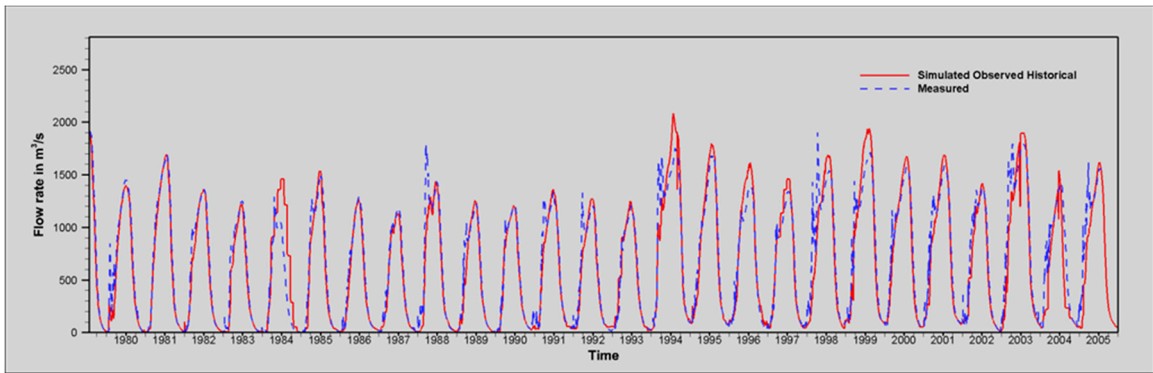

**Figure 8.** Validation of simulated historical surface water flow rates against measurements.

### 3.4. Projected Changes in Local Climate

The projected climate change scenarios are presented as relative differences for rainfall (Figure 9A) and as absolute differences for mean temperature (Figure 9B) between simulated historical and mid-century periods. All three RCMs consistently projected an increase in the mean annual rainfall, with the CANRCM4 model projecting a mean annual increase of 1.66%, followed by the WRF model projecting a rainfall increase of 1.35% and the RCA4 model with an increase of 1.05%. During the rainy season (JJA), WRF and CANRCM4 models project a large increase in rainfall, while the RCA4 project a drier future (Figure 9A,B). Similar to rainfall, mean annual temperatures are projected to increase by 1.58 °C for RCA4, 1.57 °C for CANRCM4, and 1.09 °C for WRF. For all three models, greater temperature increases are projected for the MAM period, while increases are relatively smaller for the JJA and SON periods.

### 3.5. Changes in Minimum Environmental Flow

As previously stated, the Niger River is the only permanent surface water feature from which water is continuously pumped to provide drinking and irrigation water for Niamey. Therefore, even though guidance on projected climate change impacts on river discharge is useful information, in this study, MEF flow is used as the climate change impact on surface water. The MEF is defined as the minimum flow rate of the Niger River at Niamey required to satisfy the drinking and agriculture water demand. Herein, the required minimum low flow value considered for Niamey is 55 m$^3$/s for a 10 day average period, as defined in the 2005 reference period [52]. To assess the occurrence and duration of the MEF by the end of the mid-century period (2020–2050), a python command line tool (free at GitHub https://github.com/aerler) was developed that takes as input a HGS hydrograph file, resamples it to daily average values, and counts the occurrence and duration of low/high flow.

MEF duration and occurrences are assessed under historical runoff (1980–2005), where the mid-century runoffs are kept to the historical levels (Figure 10A), and under 10% runoff reduction scenarios (Figure 10B), where runoff conditions are considered to be reduced by 10% compared to historical levels. Overall, all three models almost agree for the occurrence of the MEF, with different durations ranging from 10 to 120 days under both historical and −10% runoff scenarios.

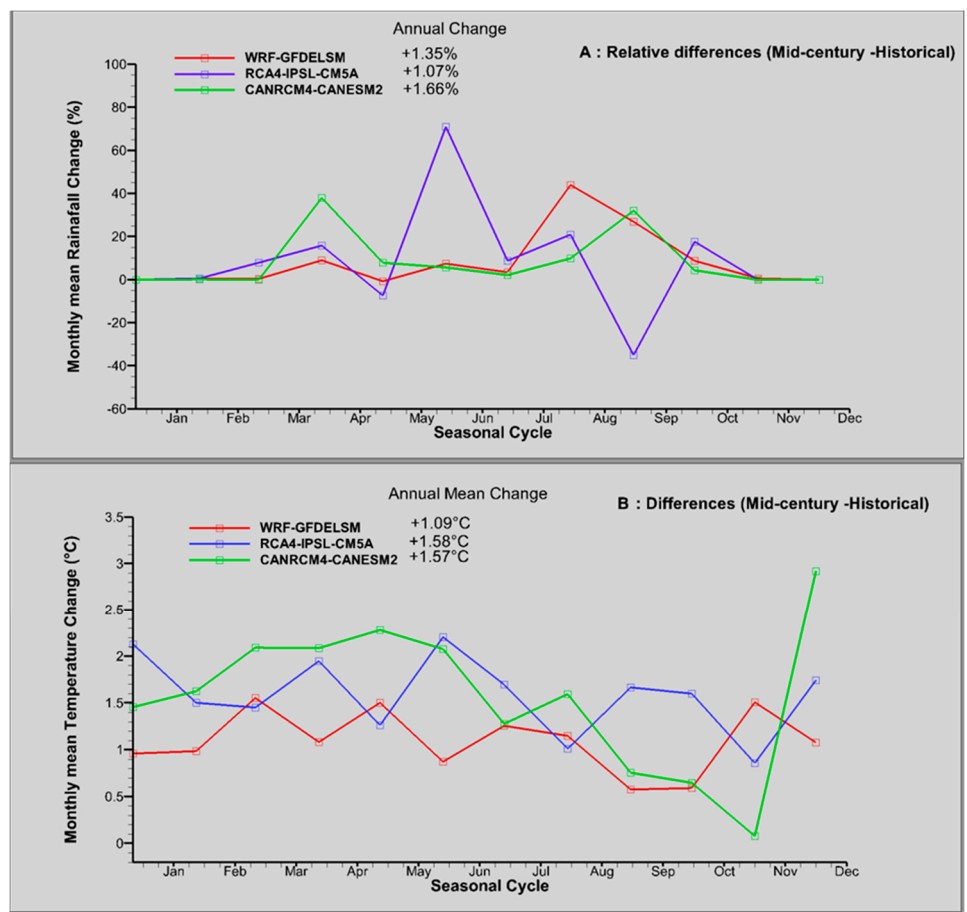

**Figure 9.** Rainfall and temperatures changes between 2020–2050 (mid-century) and 1980–2005 (historical) at Niamey Airport Station. (**A**) Relative differences (Mid-century -Historical); (**B**) Differences (Mid-century -Historical).

For the historical runoff level scenario, all three climate models agree on the MEF occurrences for the critical duration level (shown as a straight red line in Figure 10A,B), defined as a period of 10 consecutive days when the minimum flow required to satisfy drinking and irrigation water demand is not met. MEF will not be satisfied once per year from 2020 to 2025, and three times every five years between 2025 and 2035; then, occurrences will decrease to once every five years from 2035 to 2050 (Figure 10A). Conditions for MEF will be severe for the first statistical decade (2020–2030) of the mid-century, and they will become more favorable for the last two statistical decades of the mid-century. This is mostly due to the dry historical period of 1980–1990 that the Niger River experienced.

For the scenario of −10% runoff reduction, compared to the historical scenario, the occurrences of MEF are almost the same as for the historical levels with different durations. Durations of MEF are shown in Table 3 for both scenarios. The average durations of the MEF range from 53 to 76 days for 2020–2030 and between 28 to 31 days for 2020–2045, under the historical runoff scenario (Table 3). MEF average durations increase under the −10% flow reduction scenario, with durations ranging from 54 to 96 days for 2020–2030 and between 32 to 33 days for 2030–2045.

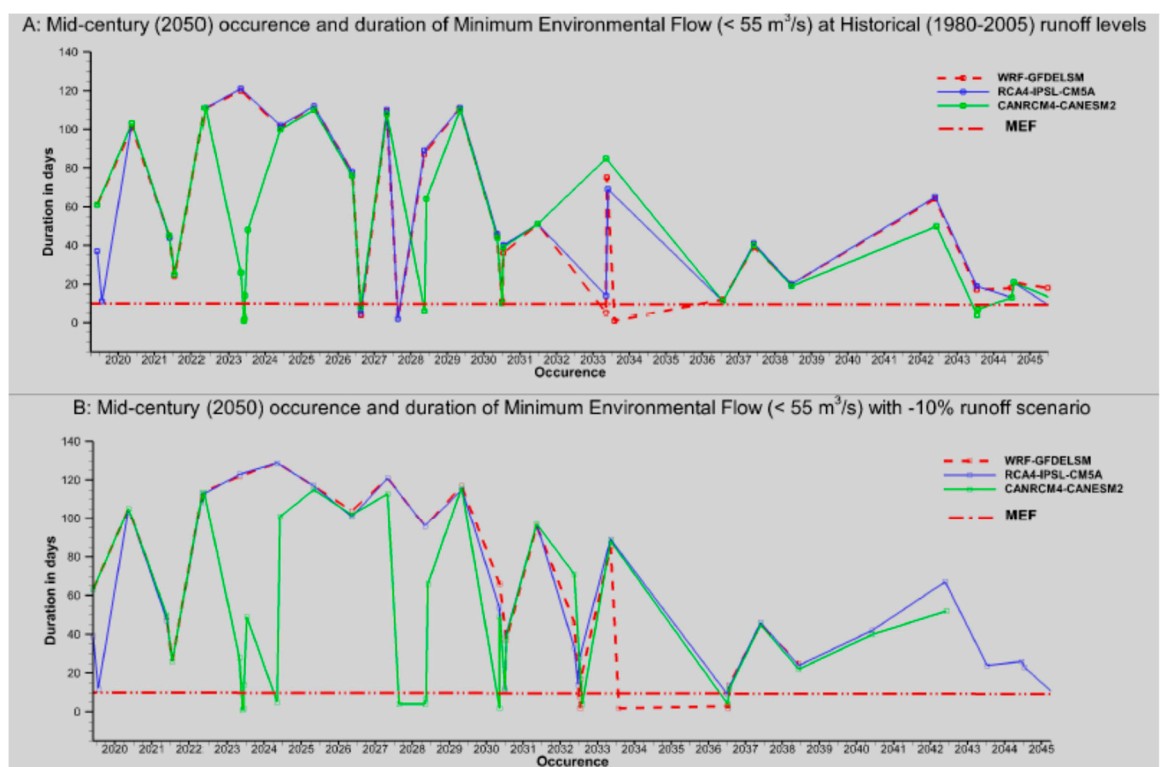

**Figure 10.** Projected occurrence and duration of minimum environmental flow (MEF) at Niamey gauging station: (**A**) runoff kept at the historical level scenario, and (**B**) scenario of −10% runoff change compared to the historical level.

**Table 3.** Duration of the MEF under historical and −10% runoff scenarios.

| Historical Runoff Scenario | Mid-Century (2020–2049) MEF Duration (days) | |
| --- | --- | --- |
| | **2020–2030 Period** | **2030–2045 Period** |
| CANRCM4 | 53 | 30 |
| RCA4 | 70 | 31 |
| WRF | 76 | 28 |
| **−10% Runoff Reduction Scenario** | **Mid-Century (2020–2049) MEF Average Duration (days)** | |
| | **2020–2030 Period** | **2030–2045 Period** |
| CANRCM4 | 54 | 33 |
| RCA4 | 88 | 32 |
| WRF | 96 | 32 |

Table 3 and Figure 10A,B show that the durations of the MEF are sensitive to runoff reduction, with projected patterns of MEF directly influenced by the changes in net precipitation of the climate scenario models. Dry (wet) models predict recurrent (less frequent) and high (low) MEF conditions by the end of mid-century. MEF conditions were previously shown to also be sensitive to runoff reduction for two gauging stations located upstream of the study area [52].

*3.6. Changes in Depth to Groundwater Table*

Figure 11 (top panel) shows the projected mean changes of the depth-to-groundwater table (Depth2GWT) for the three RCMs. All three RCMs show an increase in the groundwater table, ranging from +2 m to more than +15 m. The CANRCM4 model predicts the greater increase of groundwater table, followed by the WRF and RCA4 models. The groundwater table is likely to increase more in

high topographic areas, where the depth-to-groundwater table is deep, than topographically low areas with relatively shallow depth-to-groundwater tables. For CANRCM4 and WRF models, the mean groundwater table increased by an average of 4 m in low-altitude areas and by more than 12 m in topographically high areas. Meanwhile, the RCA4 model predicts a maximum increase of 1 m in shallow groundwater table areas and an increase of 3 m in deep groundwater table areas (Figure 11). Therefore, as seen for the MEF change patterns, groundwater table response to climate change is strongly dictated by the climate change forcing signal, particularly for mean annual net precipitation changes.

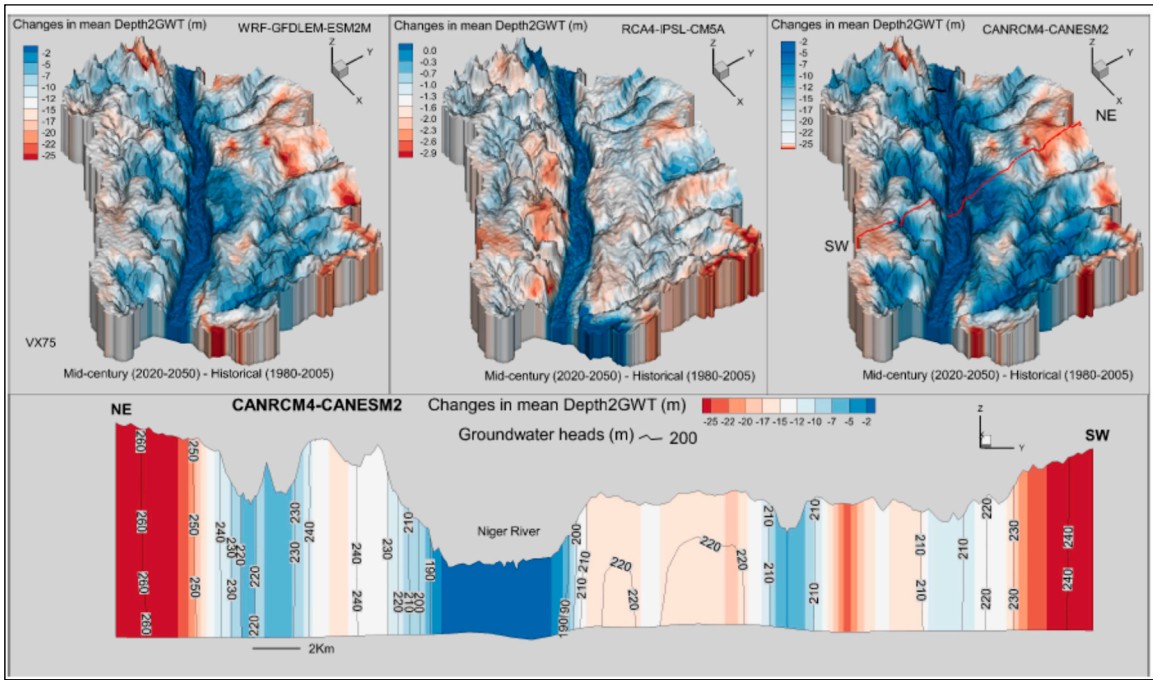

**Figure 11.** Changes in depth-to-groundwater table. Top panel: mean changes in depth-to-groundwater table by the mid-century for the three RCMs; bottom panel: two-dimensional (2D) cross-section for the CANRCM.

The groundwater table response to climate change is more perturbed in topographically high areas than in low-altitude zones (Figure 11). To illustrate the topographical effect on groundwater table response to climate change, a cross-section was drawn along the northeast (NE)–southwest (SW) axis of the study area (Figure 11, bottom). Changes in the depth-to-groundwater table for the CANRCM4 model are shown in the colored contour, and groundwater heads are represented as line contours. It is obvious that greater changes in the depth-to-groundwater table are located in areas where the groundwater heads are also greater. The topographic perturbation is still evident even within small horizontal distances (2 km). This may be explained by the intensity of seasonal variation in the groundwater table, with less variation in low topographic areas, which are groundwater discharge areas in which exchange flux mostly occurs with surface water. However, high-altitude areas are generally direct groundwater recharge areas.

The topographical perturbation of the groundwater table in response to climate change is shown in the Grand River Watershed [22]. This climate-induced topographical perturbation of the groundwater table in response to climate change appears to be significant even in small-scale topographic gradients, as shown in Figure 11. This may result from the effect of intense monsoonal rainfall inducing large seasonal groundwater table variation, as well as the local topography controlling the groundwater flow system in the study area [37,50].

*3.7. Implications of Changes in Adaptation Strategies*

Based on the projections made on surface water response to climate change impacts, the Niamey watershed in general, and the city in particular, will experience recurrent and long periods where the conditions will be under the required MEF. The risk level is increased even when the −10% runoff scenario is considered. However, the mid-century climate projection signal is wetter for all three RCMs used, which means that historical runoff level reduction is less probable. An increase in the irrigation water demand during the dry period upstream of Niamey will probably create a high surface water stress risk if the ongoing Kandadji dam construction that is supposed to maintain the MEF is not completed.

On the other hand, basin-wide groundwater table rise is projected, with greater increases for deep water tables and relatively smaller increases for shallow water tables. Therefore, considering the current population and urbanization rates, groundwater represents a sustainable adaptation pathway for the recurrent water stress that will be induced by the high sensitivity of the Niger River MEF to climate change impacts.

## 4. Summary and Conclusions

The integrated hydrological response to climate change impact has been assessed on a large-scale, semi-arid watershed using the fully integrated HGS model. The model was calibrated sequentially for long-term steady state (1980–2005), dynamic equilibrium and fully transient conditions. Three RCM models were bias-corrected at daily, monthly, and yearly timescales. Performance of the HGS simulations forced by the three bias-corrected RCMs was then evaluated against simulations forced by observed historical HGS simulations, using both surface water flow rate and depth-to-groundwater table as metrics. Mid-century (2020–2050) climate change scenario simulations were performed with the three RCMs, and the MEF and groundwater table responses were evaluated under historical and −10% runoff scenarios.

Bias correction of the historical climate scenario shows that the quantile mapping correction performed better at daily and monthly timescales than at yearly timescales. Simulations of historical depth-to-groundwater tables by the three RCMs is positively biased compared to observed historical climate simulations, with large biases at higher groundwater table depths. The resolution of the forcing RCM was found to not improve its performance significantly. All RCMs project an increase in mean annual rainfall and in mean annual temperatures. The signals of mid-century rainfall changes are directly translated into the depth-to-groundwater table response to climate change, with a general groundwater table increase predicted by the three RCMs. MEF durations show a high sensitivity to runoff reduction. The groundwater table is also found to increase more at higher altitudes than in low-altitude areas.

Fully physics-based, integrated hydrological models have been shown to represent one of the most reliable ways to assess climate change impacts on groundwater [20,22,53,54]. However, the application of these integrated models is computationally expensive, resulting in few published applications of integrated hydrological models on a large scale [53]. To the authors' knowledge, large-scale anthropogenic climate change impacts using integrated hydrological models have only three published climate change impact studies [22–24]. However, in all these studies, either small scales or relatively coarsely-meshed resolutions were used [23,24], or simulations were performed using representative seasonal cycles to reduce the computational cost [22]. Also, previously cited assessments were all performed in wet climate conditions in more developed countries, and where sufficient data are more available. Therefore, the novelty of this study is that simulations with higher resolution (up to 12 m) at a large scale (1,900 km$^2$) were performed in semi-arid climate environments with sparse data. While acknowledging the data challenge for model validations, the use of higher-mesh resolution seems to improve simulation quality, and to some extent compensates for the sparse data issue.

Development and application of fully integrated hydrological models can provide reliable guidance in addressing concerns about the combined response of surface water and groundwater to

future climate change impacts. It is therefore possible, even with modest computing resources and sparse data, to provide decision-making tools to define integrated climate change adaption strategies to water resource managers in developing countries.

**Author Contributions:** The authors contributed to this publications as follows: conceptualization, B.A.B., M.K., S.J.B. and E.A.S.; methodology, B.A.B., N.Y., S.J.B., A.R.E. and H.-T.W.; software, B.A.B., S.J.B., A.R.E., H.-T.W. and E.A.S.; validation, A.R.E., P.B. and O.S.; formal analysis, B.A.B., S.J.B. and A.R.E.; investigation, B.A.B., M.K., N.Y. and S.J.B.; resources, N.Y., S.J.B. and P.B.; data curation, S.J.B., A.R.E. and H.-T.W.; writing-original draft, B.A.B.; writing-review & editing, S.J.B., A.R.E., H.-T.W., K.S., O.S. and A.S.N.; visualization, S.J.B., A.R.E., M.K. and K.S.; supervision, A.S.N. and K.S.; project administration, N.Y.; funding acquisition, S.J.B., K.S. and E.A.S. All authors have read and agreed to the published version of the manuscript.

**Funding:** This research was funded by the German Federal Ministry of Education and Research (BMBF), grant number 01LG1202E of the WASCAL project, and Aquanty's Visiting Student Program.

**Conflicts of Interest:** The authors declare no conflict of interest.

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
