# Peer review of "High-Resolution, Integrated Hydrological Modeling of Climate Change Impacts on a Semi-Arid Urban Watershed in Niamey, Niger"

_water, doi:10.3390/w12020364_

Round 1
Reviewer 1 Report
This is a very thorough piece of work, really pointing out the care that must be taken when downscaling from GCMs to RCMs to Watershed, and with the increased need for statistical bias correction when data, particularly groundwater head data, is sparse. The expected difference, for example, between annual precipitation and evapo-transpiration historically, and that projected under future climate change, can be small, so that a projected reduction in runoff of 1.5% could be within the error of an estimate of -1.5%. I have listed some minor things below:
p. 2, lines 48-50. In the sentence that begins "Simultaneously," 3 causes for increased incidence of extreme low Niger River flows are postulated but it is not clear how each affects discharge.
p. 2, lines 54-55. May want to mention dam under construction upstream, and the need for investigation to point direction for effective climate change mitigation.
p. 2, Fig. 1a. Add scale on regional setting map. Also location of Niamey City is not shown.
p. 2. Figure 1c. Show how the three lower strata constitute the "fractured Liptaka Basement" aquifer, and are separated from the CT3 sedimentary aquifer.
p. 2, line 68, Replace "between" with "among"
p. 3, lines 101 - 106, reword to avoid repetition, perhaps beginning the second sentence of the para, "More specifically..."
p. 3, lines 110, Is it true that the study area is "located southwest of the Republic of Niger?"
p. 4, Is 2,500 mm/y actual evapotranspiration or potential?
p. 4, Fig. 2, Please put more explanation in cut-line like Annual Means from daily data, means? Also need to explain the legend and make it larger...
p. 5, lines 155 - 166, bold RDF and EDF under the equations. What is Ecan?
p. 6, lines 182 - 192, bold RA and TD under the equation. misspelled word at beginning of 192.
p. 7, line 240, should reference Fig 2
p. 7, lines 268-259, awkward sentence beginning with "The biases increase..." Also, are we discussing "biases" when we should consider this "results?"
p. 8,9, 10, Fig. 4a, b, c. Need much more explanation in the cut-line and more explanation generally in the text about how the statistical bias correction produces the results shown in Fig. 4.
p. 11, lines 330-331, Sentence beginning "The elevation..." is awkward
p. 12, line 371, delete "relatively"
p. 12, lines 372-373, What are historical groundwater point heads?
p. 12, Fig. 7, I'm not sure how encouraging the model results are, but we can definitely see the difference between Timire (screened in the CT3) and the other wells screened in the fractured bedrock.
p. 13, line 381, omit last 's' on Theses.
p. 13, line 396, sentence beginning "Table 2 shows..." does not appear to correctly describe the content of Table 2
p. 14, lines 413-414, Why are cc scenarios presented as % change for rainfall and absolute differences for mean temp?
p. 15, line 429, add 'y' to 'Niame'
p. 15, Fig. 10. Over what time interval is MEF duration calculated and plotted? Is the y-axis "Duration in days per year"? Should label the red dash-dotted line as the MEF on the chart.
p. 15, lines 450-458, this para is confusing!
p. 16, line 473, add 's' to 'station'
p. 17, Fig. 11, where does NE to SW X-section cut across the basin (CANRCM4 sim)?
p. 17, line 512, move 'MEF' to end of sentence in place of 'level.'
p. 17, line 518-519, awkward sentence, reword
p. 18, line 545, omit last 's' on 'theses'
p. 18, lines 546-553, combine sentences removing 'rarely investigated in the scientific literature.' Improve flow...this is good stuff that should be more carefully stated.
p. 18, line 554, wasn't resolution as high as 12 m in the WASCAL RCM?
Reviewer 2 Report
Broad comments
The structure of the article is correct. The authors introduce the reader into a discussed issues in detail. They provide necessary information about the research area. The authors also emphasize how few research studies have taken up the issue of climate change impact on groundwater in Africa. In my opinion, however, just mentioning previous studies is insufficient, and it would be more valuable to at least outline the results of those researches.
The methodology is clearly presented with reference to the origin. The same applies to the results. They are discussed logically and clearly. “Summary and conclusion” section describes the obtained results well.
Specific comments
The paper is written with attention to detail, so a significant inaccuracy that crept in the abstract is surprising - lines 29-30 -: "Results show that the bias correction method is optimum at daily and monthly scales...". In conclusion section - line 533-534 - one can read the opposite: "Bias correction of the historical climate scenario shows that the quantile mapping correction performed better at daily and yearly timescales than at monthly timescales.". Please, look at the figures concerning bias.
In Fig. 7, each graph contains the values of the groundwater heads from Table 2. However, there is no explanation of what they mean. I propose to either delete or describe them in the caption.
Units are missing in Tab 2.
The caption of Figure 11 is the only one in the form of a sentence. I suggest inserting dashes instead of the words "shows" twice.
There are many editing errors in the text, especially double spaces.
